

# Identification and characterization of circRNAs as competing endogenous RNAs for miRNA-mRNA in colorectal cancer

Wenliang Yuan[1,2,3,4], Sihua Peng[1,2,3], Jingyu Wang[5], Cai Wei[1,2,3], Zhen Ye[6], Ye Wang[6], Meiliang Wang[6], Hao Xu[6], Shouwen Jiang[1,2,3], Dan Sun[1,2,3], Chaoxu Dai[1,2,3], Linhua Jiang[4] and Xiaobo Li[6]

[1] Key Laboratory of Exploration and Utilization of Aquatic Genetic Resources, Shanghai Ocean University, Shanghai, China
[2] National Pathogen Collection Center for Aquatic Animals, Ministry of Agriculture of China, Shanghai, China
[3] International Research Center for Marine Biosciences at Shanghai Ocean University, Ministry of Science and Technology, Shanghai, China
[4] School of Optical-Electric and Computer Engineering, University of Shanghai for Science and Technology, Shanghai, China
[5] Department of Pathology, The First Affiliated Hospital of Jiaxing University, Jiaxing, China
[6] College of Engineering, Lishui University, Lishui, China

Corresponding author
Xiaobo Li, oboaixil@126.com

## ABSTRACT

**Background.** Recent studies showed that circRNAs are involved in the biological process of some human cancers. However, little is known about their functions in colorectal cancer (CRC).

**Methods.** Here we first revealed the expression profiles of circRNAs in the CRC tissues and the adjacent non-tumorous tissues using high-throughput sequencing. The sequence feature, chromosome location, alternative splicing and other characteristics of the circRNAs were also explored. The miRNA and mRNA expression profiles were then obtained by analyzing relevant CRC data retrived from the TCGA database. We obtained and analyzed the competing endogenous RNA (ceRNA) network of the top three pairs of the largest up-regulated and down-regulated circRNAs.

**Results.** In this study, we obtained 50,410 circRNAs in the CRC tissue and the adjacent non-tumor tissues, of which 33.7% (16,975) were new, and revealed differential changes in circRNA expression during colorectal carcinogenesis. We have identified six potential key circRNAs (circPIEZO1-3, hsa_circ_0067163, hsa_circ_0140188, hsa_circ_0002632, hsa_circ_0001998 and hsa_circ_0023990) associated with CRC, which play important roles in carcinogenesis as ceRNA for regulation of miRNA-mRNA network. In the subsequent KEGG analysis, several CRC-related pathways were found.

**Conclusions.** Our findings advance the understanding of the pathogenesis of CRC from the perspective of circRNAs and provide some circRNAs as candidate diagnostic biomarkers or potential therapeutic targets.

## INTRODUCTION

Colorectal cancer (CRC) is a common malignant tumor of the digestive system in the world (1.4 million in 2012) (*McGuire, 2016*), and more than 50% of the patients eventually die from this disease. Chemotherapy is still an indispensable treatment for CRC (*Gill, 2014*); however, with the advance of molecular biology and cell biology, targeted therapy has become a hotspot in cancer chemotherapy.

Circular RNAs (circRNAs) are a class of non-coding RNAs featuring stable structure, often showing tissue/developmental-phase specific expression (*Memczak et al., 2013*). Compared with other non-coding RNA molecules, such as miRNAs and lncRNAs, circRNAs have more desirable biomarker features, such as the stable circular structure, that can be used for disease diagnosis, for example atherosclerosis (*Burd et al., 2010*) and gastric cancer (*Li et al., 2015*). In CRC research, two recent studies demonstrated that circRNA_001569 and circular BANP modulate cell proliferation in colorectal cancer (*Zhu et al., 2017*; *Xie et al., 2016*). Recently, it was reported that hsa_circ_0020397 regulates CRC cell viability, apoptosis, and invasion (*Zhang, Xu & Wang, 2017*). *Hsiao et al. (2017)* also reported that circular RNA CCDC66 promotes colon cancer growth and metastasis.

In this study, we obtained the circRNA expression profiles of the CRC tissues and adjacent non-tumor tissues by high-throughput sequencing, and identified a small number of circRNAs with differential expression; then we analyzed the miRNA and mRNA data for CRC downloaded from the TCGA database; finally, we selected six circRNAs with the most significant differential expression to analyze their circRNA-miRNA-mRNA network. In addition, Kyoto Gene and Genomic Encyclopedia (KEGG) analyses were also performed.

## MATERIALS & METHODS

### Patients information

The CRC tissue specimens and the paired normal mucosa for circRNA detection were available from three CRC patients (two males and one female aged 58–66 years, mean age $\pm$ standard deviation (SD) 61.3 $\pm$ 4.2 years) who underwent surgery between May and October 2015 at the First Hospital of Jiaxing, China. The First Hospital of Jiaxing (Jiaxing, Zhejiang, China) granted ethical approval to carry out the study within its facilities (Ethical Application Ref: FCFHJ-2017023). All the tissues were frozen in liquid nitrogen immediately after the surgery and then stored at $-80\,°C$ until RNA extraction. All cases were newly diagnosed, histologically confirmed colorectal cancer patients, and had not received any chemotherapy or radiotherapy prior to recruitment.

### RNA Sample quality testing

We used 1% agarose gel electrophoresis to analyze the purity and integrity of the RNA. The RNA integrity number (RIN) was measured using Agilent RNA 6000 Pico Reagents (Agilent, Santa Clara, CA, USA) to assess the RNA quality. Sequencing was performed if the samples RIN values were greater than eight. The Qubit 2.0 instrument was used to accurately measure the RNA concentration.

## Sequencing library preparation and circRNA sequencing

A total amount of 1.5 μg RNA per sample was used as input material for the RNA sample preparations. Sequencing libraries were generated using NEBNext® Ultra[TM] RNA Library Prep Kit for Illumina® (NEB, Ipswich, MA, USA) following manufacturer's instructions. Then three μl USER Enzyme (NEB) was used with size-selected, adaptor-ligated cDNA at 37 °C for 15 min followed by 5 min at 95 °C before PCR. Then PCR was performed with Phusion High-Fidelity DNA polymerase, Universal PCR primers and Index (X) Primer. Finally, products were purified (AMPure XP system) and library quality was assessed on the Agilent Bioanalyzer 2100 system.

After cluster generation, the prepared libraries were sequenced on an Illumina Hiseq 4000 platform and 150 bp paired-end reads were generated.

## Screening and identification of colorectal cancer-associated circRNAs

To identify the circRNAs in the RNA-Seq data, the sequence reads were firstly mapped to the human reference genome (GRCh37/hg19, Feb., 2009) using TopHat2 (v2.1.0) (*Trapnell et al., 2012*); then, back-spliced ordering reads were extracted for circRNA predictions using CIRCexplorer (*Zhang et al., 2014*). These circRNAs were annotated by searching the circBase database (*Glažar, Papavasileiou & Rajewsky, 2014*) and the deepBase database (*Yang et al., 2009*). Finally, differentially expressed circRNAs were identified using edgeR (*Robinson, McCarthy & Smyth, 2010*), according to the criteria of a $|\log_2 FC| > 1.5$ and $P$-value $< 0.05$.

## Prediction of the potential coding ability of circRNAs

It took two steps to predict the potential coding ability of differentially expressed circRNAs through bioinformatics methods. Firstly, an online tool getorf (http://emboss. bioinformatics.nl/cgi-bin/emboss/getorf) was used to determine whether a circRNA has a open reading frame (ORF). Then, we blasted the circRNA sequences against all Internal Ribosome Entry Site (IRES) sequences using IRESite tool (*Mokrejš et al., 2009*), and the circRNAs with $E$ Value $< 0.05$ were considered to have potential encoding capability.

## Identification of differentially expressed miRNAs and mRNAs

To verify that circRNAs function as sponges or inhibitors of their interacting miRNAs, transcriptome profiling datasets were downloaded from TCGA. A data of 41 normal and 480 tumor samples for mRNA analyses were obtained. Similarly, the data of eight normal and 457 tumor samples were obtained for the miRNA analyses by the same method. Finally, the differentially expressed miRNAs and mRNAs were identified using edgeR, according to the criteria of a fold change $> 2.0$ and false discovery rate (FDR) $< 0.01$.

## miRNAs prediction, co-expression network and function analysis

The putative circRNA/miRNA interactions were investigated by miRanda (*Betel et al., 2008*) using the miRNA list from miRBase release 20.0 (*Griffiths-Jones et al., 2006*). The putative target genes of the miRNAs were predicted using the intersection of miRTarBase (*Friedman et al., 2009*) and miRDB (*Wong & Wang, 2014*). The information on the circRNAs of interest was obtained by CSCD (*Xia et al., 2018*).

The circRNA-miRNA-mRNA interaction network was constructed by Cytoscape. Cytoscape two plugins (ClueGO and CluePedia) were used for KEGG analyses, showing only the pathways with $P$-Value $< 0.05$.

## RESULTS

### Sequencing data

The sequencing yielded a total of 79.024 G of raw data, and the filtered clean data totaled 72.874 G. The quality of the sequencing data was detailed in File S1.

### General characteristics of circRNAs in CRC

A total of 50,410 circRNAs derived from 9,620 host genes were identified in the human CRC tissues and the adjacent non-tumorous tissues. Among them, 28,032 were found in circBase, 5,403 were included in deepBase, and remaining 16,975 accounting for 33.7% of the total circRNAs were observed for the first time in this study.

According to their host gene location, the 50,410 circRNAs were widely distributed on all the chromosomes (Fig. 1A). Specifically, only chromosome 1 and chromosome 2 produced more than 4,000 circRNAs. Most of the other chromosomes generated more than one thousand circRNAs, except chromosome 21, Y and chrUn (with 542, 81 and 3 circRNAs, respectively). Our data showed that 49,801 (98.8%) circRNAs were excluded from the first or last exons of their host genes (Fig. 1B). In addition, we found that about 66.5% of the host genes produced multiple circRNA isoforms (Fig. 1C). We found that the BIRC6 host gene produced the highest numbers of circRNAs isoforms. Interestingly, it was described in other studies that BIRC6 over-expression is a predictor of poor prognosis in CRC (*Hu et al., 2015*). Most exonic circRNAs consisted of multiple exons, with the most circRNAs containing two or three exons, and the maximum number of exons in a circRNA was 48 (Fig. 1D).

### Screening of the differentially expressed circRNA

The differentially expressed circRNAs between the CRC tissues and the adjacent non-tumorous tissues were identified. Finally, 98 circRNAs were identified, of which 49 were up-regulated and 49 were down-regulated (File S2). The hierarchical clustering (Fig. 2A) and volcano plots (Fig. 2B) showed the variation of circRNA expression between the normal and the CRC samples. Additionally, the host genes of these differentially expressed circRNAs were derived from exonic regions (94), intronic regions (1, circMYO7B-3) (Fig. 2C), etc.

To predict the potential coding ability of the differentially expressed circRNAs, we found that 69 circRNAs (70%) contained at least one ORF, but only eight circRNAs had IRESs (Fig. 2D). To investigate the functional association of the host genes of the differentially expressed circRNAs in CRC, we analyzed the genes using the GeneMANIA plugin in the Cytoscape software (Fig. 2E). Most of the network interactions were co-expression, physical interactions and genetic interactions. The complex interaction between host genes suggests that this correlation may also exist between differentially expressed circRNAs.
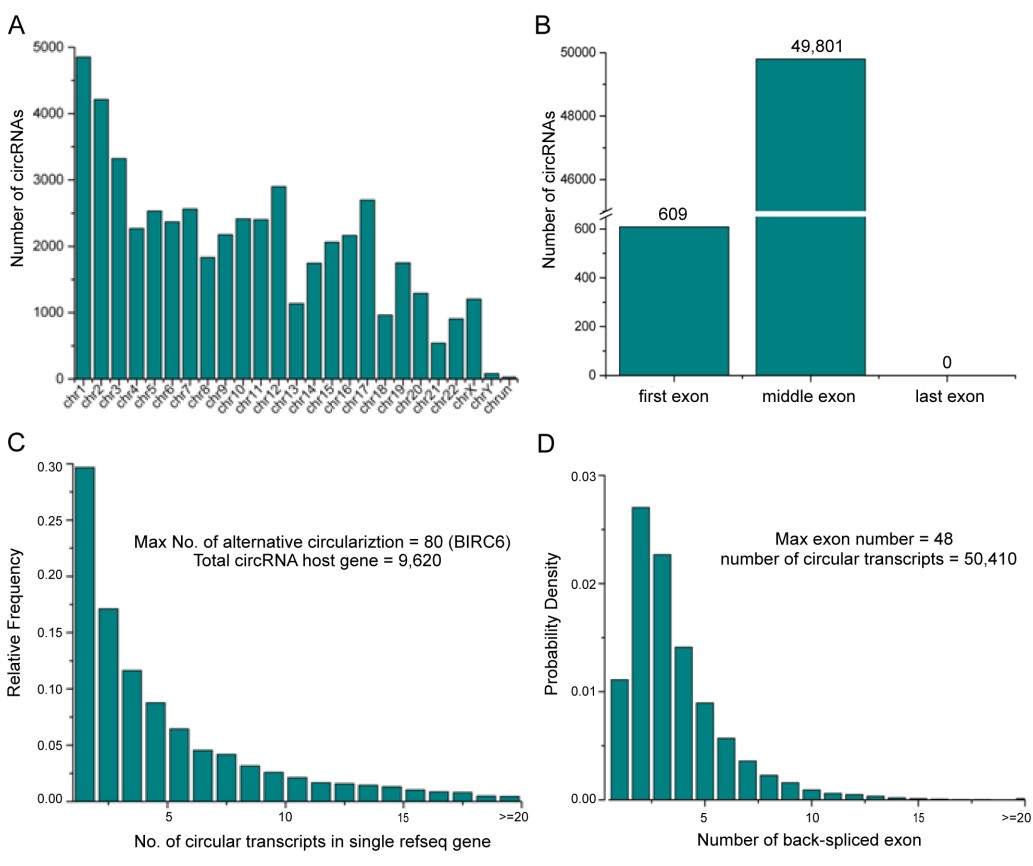

**Figure 1 General characteristics of circRNAs in CRC.** (A) Genomic features of circRNAs expressed in human CRC. Chromosomal distribution of the circRNAs. (B) Distribution of the back-spliced exons in circRNAs. (C) Distribution of the number of different types of circRNA transcripts from each circRNA host gene. (D) Distribution of the number of back-spliced exons in each circRNA.

## Screening of differentially expressed miRNAs and mRNAs

According to the criteria of $|\log_2\text{FC}| > 2$ and $q$-value $< 0.01$, 245 pre-miRNAs (DE_pmiRNA) and 2,083 mRNAs (DE_mRNA) were identified as aberrantly expressed in the CRC tissues compared with the adjacent non-tumorous tissues (Files S3 and S4). It was found that many miRNAs and mRNAs were up-regulated or down-regulated more than 100-fold (Figs. 3A–3B).

## Interaction between differentially expressed circRNAs, miRNAs and mRNAs

Evidence showed that circRNAs function as sponges or inhibitors of their interacting miRNAs to terminate regulation of their target genes (*Zhang, Xu & Wang, 2017*; *Hsiao et al., 2017*). We obtained 1,666 pre-miRNAs including binding sites of the differentially expressed circRNAs, and then obtained 3,707 target genes of these pre-miRNAs by searching the three databases. Furthermore, by analyzing DE_pmiRNA and these pre-miRNAs, we obtained 192 miRNAs in the intersection, so these miRNAs can interact with the circRNAs (Fig. 3C). Similarly, we obtained 225 DE_RNAs related to the differentially expressed
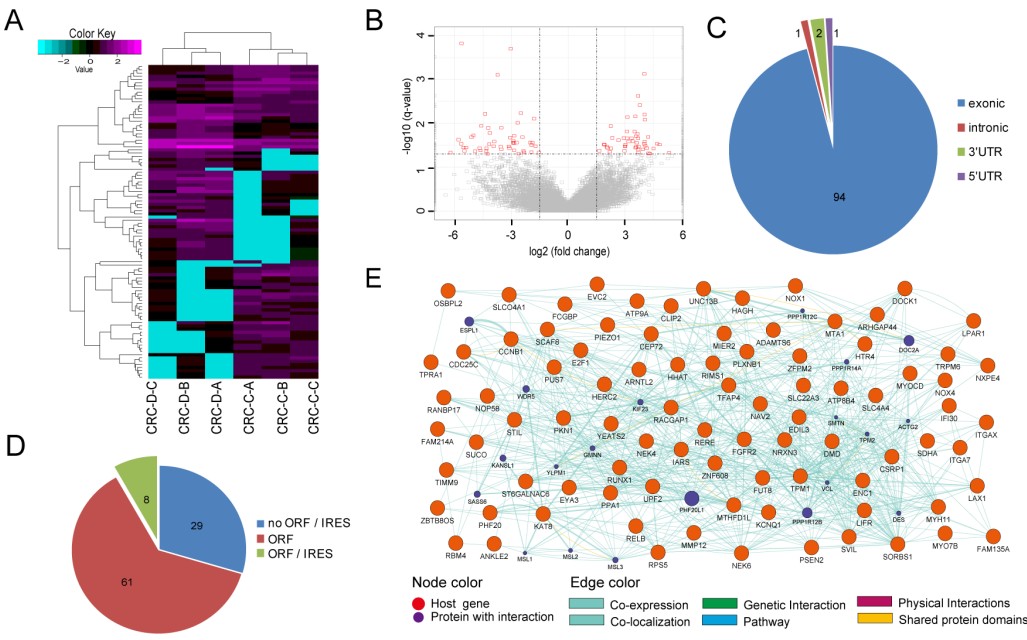

**Figure 2** **Differential expression of circRNAs in CRC tissues.** (A) Hierarchical clustering analysis of the circRNAs. CRC-D-A, CRC-D-B and CRC-D-C are adjacent normal tissue samples. The remaining three are cancer tissue samples. (B) Volcano plots are constructed using the fold-change values and *q*-values. The red dots in the figure represent statistically significant differentially expressed circRNAs. (C) Distribution of genomic regions that differentially expressed circRNAs derived from: exonic, intronic regions, etc. (D) Potentially encoded protein analysis of differentially expressed circRNAs. (E) GeneMANIA network of host genes of differentially expressed circRNAs.

circRNAs (Fig. 3D), and in this process, 40 DE_pmiRNAs were discarded because their target genes did not appear in this set (Fig. 3E). Interestingly, even if only 123 DE_pmiRNAs were retained, all of the differentially expressed circRNAs were still retained.

## Networks regulated by circRNAs

We selected the top three down-regulated (circPIEZO1-3, hsa_circ_0067163, and hsa_circ_0140188) and up-regulated circRNAs (hsa_circ_0002632, hsa_circ_0001998 and hsa_circ_0023990) as the hub components referring to recent studies (*Gargouri et al., 2015*). As shown in Fig. 4A, we found that all the six circRNAs belonged to the exonic circRNA and were all cyclized by multiple exons. We also found that the expression of hsa_circ_0140188 was significantly down-regulated, and the expression of its host gene DMD was decreased. Similarly, hsa_circ_0023990 and the host gene NOX4 were highly expressed. However, this consistent change in expression did not occur in the remaining four cirRNAs and their host genes, probably because circRNAs have a higher stability.

To investigate the potential mechanisms of circRNA in the development and progression of CRC, we constructed the circRNA-miRNA-mRNA interaction network for these six circRNAs. The ceRNA interaction network consists of six circRNAs, 35 DE_miRNAs and 64 DE_mRNAs (Fig. 4B). By querying the clinical data in the TCGA database, we found that the expression levels of the six ceRNA-related mRNAs significantly correlated to

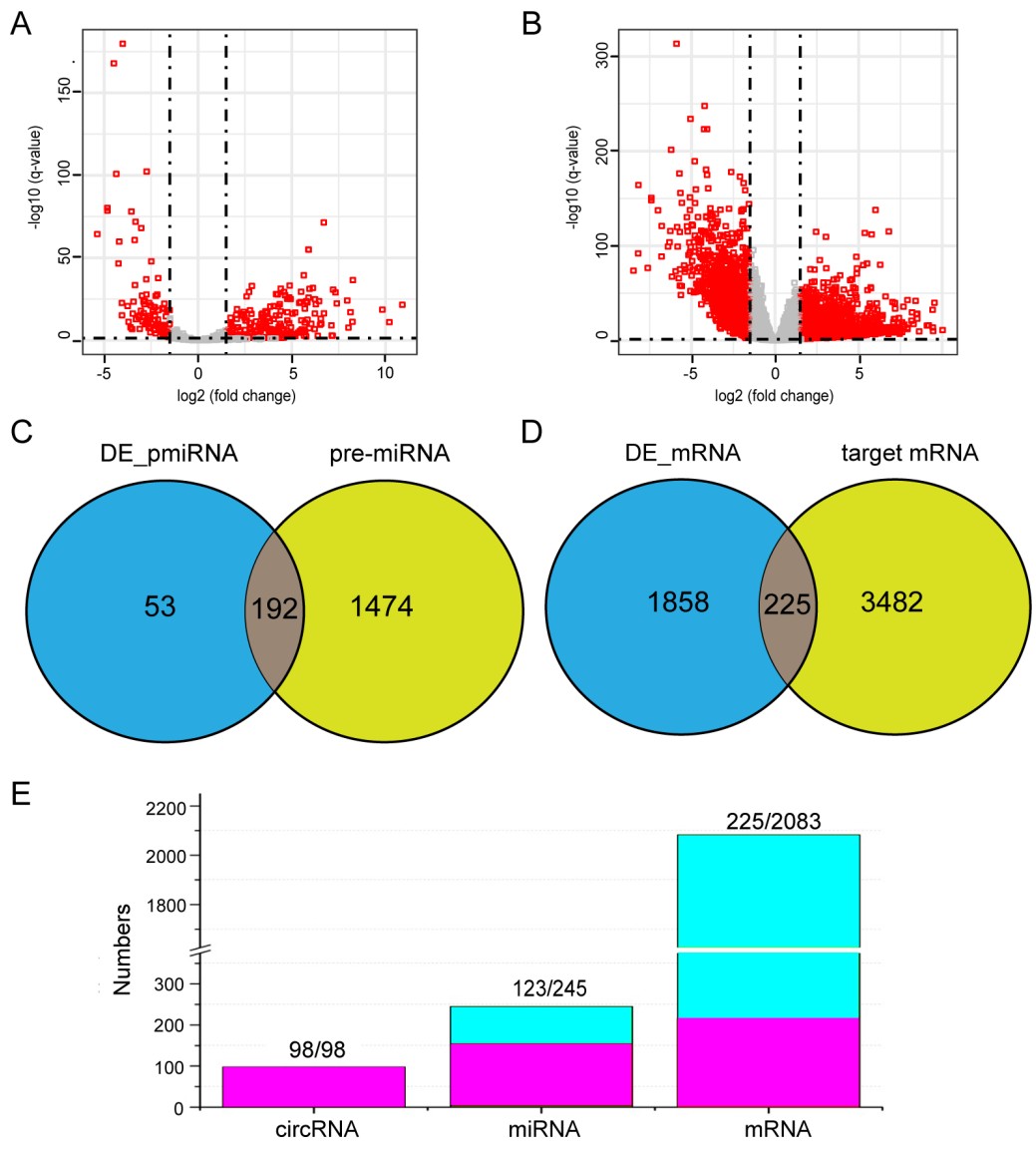

**Figure 3** **Differential expression analysis and interaction analysis of miRNAs and mRNAs.** Volcano plots showing expression profile of pre-miRNAs (A) and mRNAs (B). (C) The intersection of the differentially expressed pre-miRNAs (DE_pmiRNA) and 1,666 pre-miRNAs. (D) The intersection of the differentially expressed mRNA (DE_mRNA) and 3707 miRNA target genes. (E) Comparison of data sizes before and after data processing. Purple indicates retained data and blue indicates discarded data.

the survival time of the CRC patients (Fig. 4C), suggesting that circRNAs-selected may have prognostic value. We found that high expression of hsa_circ_0023990 significantly improved the survival time in the patients with CRC due to the high expression of SOX1, AQP6 and ITGBL1. Similarly, low expression of hsa_circ_0067163 correlates to a poor survival due to low expression of TPM2.

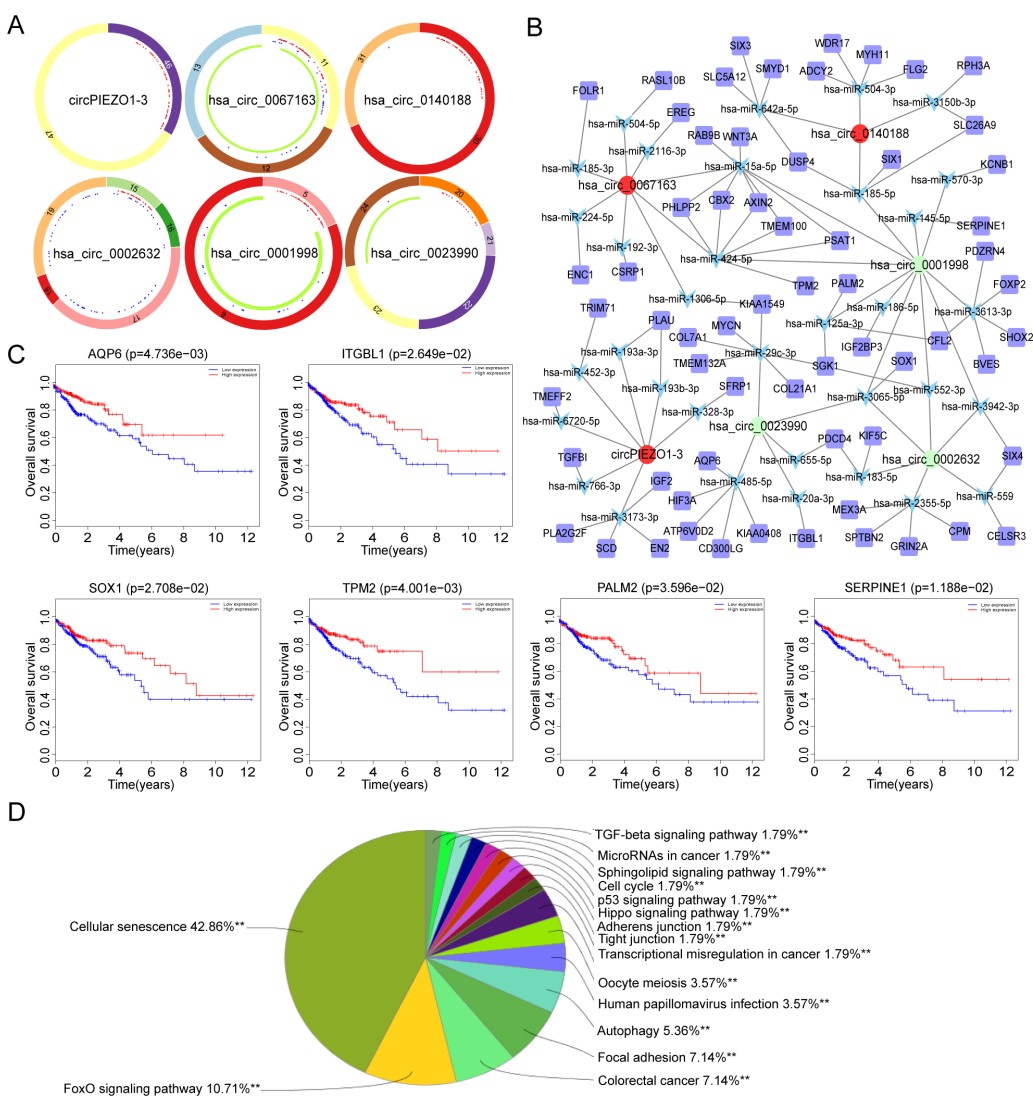

**Figure 4** **Information on six hub circRNAs.** (A) The top three up-regulated and down-regulated circRNAs. The outer loop represents the exon that constitutes the circRNA, the innermost green ring represents the ORF, the middle red triangle represents the microRNA response element, and the blue cross point represents the RNA binding protein. (B) CircRNAs-miRNAs-mRNAs network. The red circle denotes the down-regulated circRNAs, the green circle denotes the up-regulated circRNAs, the blue inverted triangle denotes miRNAs, and the purple rectangle denotes the mRNAs. (C) Survival analysis of ceRNA—associated mRNA. (D) The KEGG pathway analysis of the top three circRNA pairs in the up-regulation and down-regulation circRNAs.

## Functional enrichment analysis of circRNAs

The functional roles of most circRNAs have not been characterized; however, it would be beneficial to predict signaling pathways involving circRNAs by bioinformatics methods. Therefore, according to the obtained ceRNA network and the target genes of the miRNA in the network, the KEGG pathway analysis of the six circRNAs was performed (Fig. 4D). There were 17 KEGG pathways significantly enriched in our study ($P < 0.01$). Among
these pathways, some were directly linked to cancer pathogenesis, such as colorectal cancer, p53 signaling pathway (*Slattery et al., 2018*), TGF-β signaling pathways (*Xu & Pasche, 2007*) and microRNAs in cancer. Interestingly, although other pathways, such as cellular senescence (*Schmitt, 2007*) and Foxo signaling pathway (*Liang et al., 2017*), seemed not to be directly related to CRC, they were also found associated with the development of multiple diseases.

## DISCUSSION

Up to now, many circRNAs have been found in various human normal or diseased tissues. Researchers have identified 8,045 in heart, 3,982 in liver (*Zheng et al., 2016*), 15,996 in testis (*Dong et al., 2016*) and 65,731 in normal human brain (*Rybak-Wolf et al., 2015*). In our study, we predicted 50,410 circRNAs in the normal and diseased human colorectal tissues. Compared with other organs (for example heart, liver and testis), the expression of the circRNAs in the human colorectal tissues are the most abundant. Our data showed that most circRNAs are excluded from the first or last exons of their host genes, which is consistent with previous research that back-spliced events are generally difficult to occur in the first or last exons of the host genes (*Zhang et al., 2014*). In the present study, we found that 66.5% of the 9,620 host genes produce multiple circRNA isoforms, suggesting that there are other factors contributing to the occurrence of back-spliced events, for example, non-repetitive sequences are largely included in these "hot-spot" genes (*Schmitt, 2007*).

Research showed that most circRNAs are derived from exonic regions and 5′ UTR sequences (*Rybak-Wolf et al., 2015*). Data analysis of our differentially expressed circRNAs also supports this view. Recent studies showed that circRNAs directly translate proteins and participate in various physiological processes (*Yang et al., 2017*; *Legnini et al., 2017*). We analyzed the differentially expressed circRNAs and found that most of them contained ORF and IRES, indicating that these circRNAs have potential coding ability.

As is known, some oncogenes, such as RNA binding protein, ribosomal protein S5 (RPS5) and 5-hydroxytryptamine receptor 4 (HTR4), are differentially expressed in CRC compared with adjacent normal tissues (*Shimoyama et al., 2016*; *Hou et al., 2018*). In our differentially expressed circRNAs, the host genes of hsa_circ_0128314 and hsa_circ_0005598 are HTR4 and RPS5, respectively. Therefore, we believe that some oncogenes will not affect their carcinogenic properties even if they are cyclized during transcription.

CeRNA hypothesis describes the mechanism for a class of RNAs with miRNA binding sites that competitively bind to miRNAs to inhibit their regulation of the target genes (*Tay, Rinn & Pandolfi, 2014*; *Salmena et al., 2011*). The carcinogenic mechanism of circRNAs may occur through their miRNA-mediated effects on the gene expression, as circRNAs have more miRNA binding sites and are highly stable (*Wilusz & Sharp, 2013*; *Guo et al., 2014*). In our study, based on the ceRNA hypothesis, we utilized paired circRNA, miRNA and mRNA expression profiles of the CRC patients combined with experimentally validated miRNA-target interactions to reconstruct circRNA-associated ceRNA network for the progression of CRC. However, Our findings are preliminary and have some limitations because our findings were only based on bioinformatics analyses and extensive wet-lab

validation experiments are needed. As for our future experimental validation plan, we will focus on the *in vitro* validation of differential expression of hsa_circ_0023990 to verify its correlation with differentially expressed multiple mRNAs, such as SOX1, AQP6 and ITGBL1.

In the ceRNA network of the selected "hot-spot" circRNAs, we found that some miRNAs have been confirmed to promote colorectal cancer pathogenesis for their expression difference by other studies, such as hsa-miR-29c-3p (*Chen et al., 2017*), suggesting that circRNA plays a role in the development of cancer by absorbing functional miRNAs to regulate the expression of corresponding genes. In addition, we also found some more complex regulatory relationships between circRNAs and miRNAs, for example, high expression of hsa_circ_0067163 and low expression of hsa-circ_0001998 simultaneously acted as a "sponge" of hsa-miR-424-5p, which leaded to low expression of its target gene TPM2.

The occurrence of colorectal cancer is not simply caused by a single signal pathway. Its occurrence and development are the result of the accumulation of multiple signal pathways, which are regulated by the network interlaced downstream of the pathway. Abnormalities in each pathway may cause disorder and/or cause colorectal cancer. The TGF-β signaling pathway regulates cell proliferation, differentiation, migration, apoptosis, and regulates stem cell repair (*Fleming et al., 2012*). The transcriptional co-activator with PDZ-binding motif and Yes-associated protein integrates with Wnt and TGF-β signaling in several cells and may have a significant effect on intestinal cell proliferation, differentiation and other functions (*Xu & Pasche, 2007*).

## CONCLUSIONS

In summary, in this study, we obtained 50,410 circRNAs in the CRC tissue and the adjacent non-tumor tissues, of which 33.7% (16,975) were new, and revealed differential changes in the circRNA expression during colorectal carcinogenesis. We have identified six potential key circRNAs associated with CRC, which play important roles in carcinogenesis as ceRNA for the regulation of the miRNA-mRNA network. Our findings advance the understanding of the pathogenesis of CRC from the perspective of circRNAs and provide some circRNAs as candidate diagnostic biomarkers or potential therapeutic targets.

## ACKNOWLEDGEMENTS

We thank Dr. Xiaoning Peng and Dr. Zuozhou Chen for their helpful suggestions.

### Funding

This work was supported by the National Natural Science Foundation of China (61373057 to Xiaobo Li, 61775139 to Linhua Jiang), the Zhejiang Provincial Natural Science Foundation of China (LY17F020003 to Xiaobo Li), the Zhejiang Provincial Key Laboratory of Digital Design and Intelligent Manufacturing of Characteristic Cultural and

Creative Products (2016E10007 to Xiaobo Li), the Shanghai Natural Science Foundation (15ZR1420800 to Sihua Peng), and the Jiaxing Municipal Science and Technology Project (2015AY23012 to Jingyu Wang). There was no additional external funding received for this study. The funders had no role in study design, data collection and analysis, decision to publish, or preparation of the manuscript.

## Grant Disclosures

The following grant information was disclosed by the authors:
National Natural Science Foundation of China: 61373057, 61775139.
Zhejiang Provincial Natural Science Foundation of China: LY17F020003.
Zhejiang Provincial Key Laboratory of Digital Design and Intelligent Manufacturing of Characteristic Cultural and Creative Products: 2016E10007.
Shanghai Natural Science Foundation: 15ZR1420800.
Jiaxing Municipal Science and Technology Project: 2015AY23012.

## Competing Interests

The authors declare there are no competing interests.

## Author Contributions

- Wenliang Yuan performed the experiments, analyzed the data, prepared figures and/or tables, authored or reviewed drafts of the paper, approved the final draft.
- Sihua Peng conceived and designed the experiments, performed the experiments, analyzed the data, contributed reagents/materials/analysis tools, prepared figures and/or tables, authored or reviewed drafts of the paper, approved the final draft.
- Jingyu Wang, Shouwen Jiang and Chaoxu Dai performed the experiments, approved the final draft.
- Cai Wei performed the experiments, analyzed the data, contributed reagents/materials/analysis tools, prepared figures and/or tables, approved the final draft.
- Zhen Ye, Ye Wang and Meiliang Wang performed the experiments, contributed reagents/materials/analysis tools, approved the final draft.
- Hao Xu performed the experiments, contributed reagents/materials/analysis tools, prepared figures and/or tables, approved the final draft.
- Dan Sun performed the experiments, analyzed the data, approved the final draft.
- Linhua Jiang and Xiaobo Li conceived and designed the experiments, authored or reviewed drafts of the paper, approved the final draft.

## Human Ethics

The following information was supplied relating to ethical approvals (i.e., approving body and any reference numbers):

The First Hospital of Jiaxing (Jiaxing, Zhejiang, China) granted Ethical approval to carry out the study within its facilities (Ethical Application Ref: FCFHJ-2017023).

## Data Availability

The RNA-seq data are available at NCBI BioProject (ID: PRJNA521856).
## Supplemental Information

Supplemental information for this article can be found online at http://dx.doi.org/10.7717/peerj.7602#supplemental-information.

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
