# Peer review of "Identification and characterization of circRNAs as competing endogenous RNAs for miRNA-mRNA in colorectal cancer"

_PeerJ, doi:10.7717/peerj.7602_

## Round 0.1 · original submission · Major Revisions

Please address the concerns raised by the reviewers.

Reviewer 1 ·

Basic reporting

Yuan W et al have identified some circRNAs as putative candidate diagnostic biomarkers or potential therapeutic targets for colorectal cancer (CRC). To achieve this, they compared the expression profiles of circRNAs in CRC and the adjacent non-tumorous tissues using high-throughput sequencing. They found 50,410 circRNAs in the CRC tissue and the adjacent non-tumor tissues, of which 33.7% (16,975) are new. Interestingly, they also uncovered that six circRNAs associated with CRC could play important roles in carcinogenesis as competing endogenous RNA (ceRNA) for regulation of miRNA-mRNA network. However, to make the story more compelling, the authors need to provide additional information and validations that could directly test their model.

Specific comments:

1) Figure 1C: you indicate in this figure that the BIRC6 host gene produces the highest numbers of circRNAs isoforms. Interestingly, it was described in other studies that BIRC6 overexpression is a predictor of poor prognosis in colorectal cancer. What about your data (mRNA and miRNA differentially expressed, circRNAs…), could you discuss about this point?

2) About the previous point, could you also please include in your study an excel file that contains the lists of differentially expressed miRNAs and mRNAs?

3) The current phrasing in your paragraph “Interaction between differentially expressed circRNAs, miRNAs and mRNAs” (lane 177 to 186) makes comprehension difficult. Could you make it easier to understand? Especially when we compare the different numbers shown in Figures 3C, 3D and 3E with the numbers mentioned in this paragraph. In addition, could you also add an Excel file containing the list of the mature miRNAs and the list of mRNAs?

4) I suggest that you improve the description and the discussion relative to the figures 4B and 4C to provide more justification for your study (negative/positive survival factors…). For example, is a low expression of hsa_circ_0067163 associated with a poor survival due to a low expression of TPM2 and high expression of has-miR-424-5p? For the other examples, it seems that the other circRNAs are up-regulated which in return should increase the expression level of the target genes and thus the survival curves.

5) Finally, could you also validate in cells your model about the Networks Regulated by circRNAs by using an overexpression strategy and/or a RNA interference strategy with siRNA or shRNA directed to the back-splicing site of the different circRNAs and then quantify the expression levels of the miRNAs and target mRNAs?

Minor comments:

L49: replace “More” by more.
L72-73: the text is highlighted in yellow.
L79: are you sure about the bioanalyzer chips RNA 7500 or did you used RNA pico or nano Chips?
L98: the base of the logarithm is missing in your formula.
L113: replace “miRNAsprediction” by miRNAs prediction.
L132: only chromosome 1 and chromosome 2 produced more than 4,000 circRNAs.
L146-148: you need to rephrase the sentence.
L152: replace “we foundthat” by we found that.
L160: in the description of the figure 2B you indicate p-values while in the volcano plots you mentioned q-value (the same for figure 3A and 3B).
L169: instead of “(Fig. 3)”, you have to precise Fig 3A and 3B.
L173: miRNAs or pre-miRNAs?
L222: instead of brain (65,731) it should be testis (15,996).
L223: Since in normal brain tissue it was identified 65,731 circRNAs and you identified 50,410 circRNAs in colorectal tissues, you should change your statement about “most abundant” in your sentence.
L240: Competing endogenous RNA (ceRNA). You have to indicate the definition of ceRNA at the lane 35 when you used it for the first time in your text.
L249: I did not find hsa-mir-96-5p in your ceRNA network of the selected "hot-spot" circRNAs (Figure 4B).
Figure 1C: replace “Relatuve Frequency” by Relative Frequency.
Figure 2C and lane 159: replace “CPC tissues” by CRC tissues.
Figure 2C: replace “exontic 94” by exonic 94.
Figure 3D: replace “targe mRNA” by target mRNA.

Experimental design

As mentioned in the previous section, supplemental data/figures are required to strengthen the manuscript.

Validity of the findings

As mentioned in the previous section, supplemental data/figures are required to strengthen the manuscript.

Reviewer 2 ·

Basic reporting

no comment

Experimental design

no comment

Validity of the findings

no comment

Additional comments

Comments to the authors

In general, aberrant expressions of genes are observed in cancers. Circular RNAs (circRNAs) are known to be generated by splicing of precursor messenger RNAs (pre-mRNAs). Therefore, it is assumed that featured circRNAs exists in each cancer. Indeed, the authors have found 98 featured circRNAs in Colorectal cancer (CRC) in this manuscript (Supplemental file 2). If the featured circRNAs function as sponges or inhibitors of their interacting microRNA (miRNA), the miRNA-targeted mRNAs would be controlled through the expression level of the featured circRNAs in CRC, probably resulting in malignant cell transformation in this cancer. Based on this idea, the authors have attempted to find the circRNAs-miRNAs-mRNA network in CRC. However, the data are not enough to demonstrate the network in this manuscript. In particular, explanations for the data are totally insufficient to inform the author’s thought based on the data. Several concerns are enumerated below.

1. Figure 2A – There is no explanation about CRC-D-C, B, A and CRC-C-A, B, C. The author should clear which lanes are the normal samples or the CRC samples.

2. Figure 2E – The lines between the circles are illegible. The authors should describe the relationship between this network analysis and the CRC.

3. Figure 3C and D – What is “DE_” ? What is pre-miRNA? What is target mRNA? Nothing explains about them in detail in Figure legends. It is difficult to understand the description of this data on lines 178-186. The authors would show the flow chart concerning this analysis.

4. Figure 4A – What are host genes of these circRNAs? Are the host genes included in the differential expression of mRNAs between the normal and the CRC samples?

5. Figure 4C – What is different between blue and red lines? The authors should describe this point in Figure legends.

6. Finally, the authors should perform some wet experiments to confirm their dry analysis. For instance, they could perform the luciferase reporter experiments using the reporter gene bearing the binding sites of miRNA which could interact with a specific circRNAs in CRC. These experiments would be useful for confirming the circRNAs-mRNAs-mRNA network in CRC.

Minor point
7. line 151 – Is Fig. 1C correct? The author should check this point.

Reviewer 3 ·

Basic reporting

No comment

Experimental design

No comment

Validity of the findings

Yuan et al. have found various misregulated circRNAs in colorectal tissue samples impacting the expression level of hundreds of miRNAs and mRNAs. Among the 6 top-misregulated circRNAs, they have shown that a regulation network (as ceRNA) is involved in pathogenesis in CRC. These data suggest that those 6 circRNAs could be used as potential diagnostic biomarkers.
However, authors's conclusion are not fully supported by direct evidence and need more decisive experiments.

Major Points:

1) It will be interesting to test if one of the top 6 deregulated circRNAs act as a miRNA sponge. Hsiao et al. (Cancer Research, 2017) have shown that circCCDC66 act as miRNA sponge for specific miRNAs with a nice experiment.

2) Can you analyse the expression level of miRNAs interacting with circRNA by RT-qPCR and also mRNAs targeted by those miRNAs still by RT-qPCR with and without siRNA against your top misregulated circRNAs?

Minor Points:

1) Line 49: change More by more
2) Line 113: miRNA prediction in the title
3) Line 152: found that
4) Line 153: circ instead of cicr
5) Fig 1C Relative Frequency
6) Fig 2C Exonic
7) Can the authors test the effect of downregulation of upregulated circRNA (with siRNA) on CRC proliferation with MTT assay (or another one)

---

## Round 0.2 · Minor Revisions

Thankyou for making the necessary modifications to your manuscript. There are still a few outstanding issues highlighted by the reviewers. Please make these modifications to the manuscript.

Furthermore, the lack of data validation is still an outstanding issue. You should more clearly state the limitations of this study and note that the findings are preliminary (only done in one set of samples) throughout the manuscript.

Reviewer 1 ·

Basic reporting

the authors took notice of the recommendations and also provided us supplementary files requested.

However, it could have been great if they performed some wet lab validations as requested by all the reviewers. For example, to validate the new circRNAs but most importantly to validate the circRNAs- miRNA-mRNA network (these validations could have been done at least in cell lines).

Moreover, I totally understand the point raised by the authors concerning the grants/funding issues… It is complicated for all of the scientific community. However, even if this manuscript is “only” a preliminary study and the fact that other journals allow that, it is not an argument. Indeed, we are talking about science.

I appreciate that the authors described the shortcomings of this manuscript in the discussion section. However, in order to avoid misleading the reader and without wet lab validations of the model, they should have done that since the beginning by talking about correlative study, putative diagnostic biomarker or which could play important roles…

Overall, this manuscript has still a good potential:
- circRNAs profiling in CRC and the adjacent non-tumorous tissues using high-throughput sequencing.
- miRNA and mRNA expression profiles retrieved from the TCGA database.
- the correlative study (circRNAs- miRNA-mRNA network).

Specific comments:

• in PDF, the resolution of the figures is bad in particular when we try to zoom in.
• L79: size of the font is different.
• L111: replace “miRNA sprediction” by miRNAs prediction.
• L180 -188: I suggest that you improve the description of this paragraph (Interaction between differentially expressed circRNAs, miRNAs and mRNAs) to make it easier to understand.
• L197: you should verify but it seems that only 3 (instead of 6) ceRNAs related mRNAs were significantly correlated with the survival time of CRC patients. It is true that several parameters can regulate the expression level and/or the stability of miRNAs, that's why some wet lab experiments should have been done to validate your network. For example, when the expression of hsa_circ_ 0023990 is high, the survival time is improved in patients due to high expression of SOX1. Unfortunately, the expression level of the corresponding miRNA (hsa-miR-3065) is also high. The same for ITGBL1 and hsa-miR-20a.
• L201: you should replace “associated with” by “correlates with”

Experimental design

the recommendations are mentioned in the previous section

Validity of the findings

the recommendations are mentioned in the previous section

Reviewer 2 ·

Basic reporting

no comment

Experimental design

no comment

Validity of the findings

no comment

Additional comments

Comments to the authors

The authors have answered in a satisfactory manner to raised issues in most of the comments. While the quality of manuscript has largely improved, a few issues still remain unclear. The concerns are enumerated below.

1. Figure 4A – The top 3 down-regulated circRNAs (circPIEZO1-3, hsa_circ_0067163 and hsa_circ_0140188) in CRC are derived from PIEZO1, TPRA1 and DMD genes, respectively, whereas the 3 up-regulated circRNAs (hsa_circ_0002632, hsa_circ_0001998 and hsa_circ_0023990) come from STIL, FUT8 and NOX4 genes, respectively. Expectedly, the analysis of the differentially expressed mRNAs in the CRC shows that the expression of DMD gene is decreased, while that of NOX4 gene is elevated in the CRC (additional_file_4), suggesting that the production of hsa_circ_0140188 and hsa_circ_0023990 is due to the expression amount of the host genes. This is a reasonable event. On the other hand, the other host genes (PIEZO1, TPRA1, STIL and FUT8) do not include in the list of the differentially expressed mRNAs in the CRC (additional_file_4). Therefore, the authors should explain how to alter the expression of circPIEZO1-3, hsa_circ_0067163, hsa_circ_0002632 and hsa_circ_0001998 in the CRC.

2. Figure 4B – The authors mentioned that low expression of hsa_circ_0067163 associated with a poor survival due to a low expression of TPM2 via the function of hsa-miR-424-5p in the new version of the manuscript on lines 242-243. However, the hsa-miR-424-5p is also targeted by hsa-circ_0001998 which is one of the up-regulated circRNAs in the CRC (Figure 4B). Therefore, it is unclear why the expression of TPM2 is decreased in the CRC. The authors should describe this discrepancy concerned with the sponge effect of hsa_circ_0067163 and hsa-circ_0001998 to hsa-miR-424-5p in the CRC.

Minor point
3. Figure 4C – It is still unclear what is blue and red lines. There is no description regarding the lines in the figure legends of the new version of the manuscript. The annotation of the lines is too small in the graph.

Reviewer 3 ·

Basic reporting

Authors should bring some minor modifications to their article:

line 47: Change "diseasee" by disease
line 111: correct the title of paragraph

Experimental design

No comment

Validity of the findings

No comment

Additional comments

Authors should bring some minor modifications to their article:

line 47: Change "diseasee" by disease
line 111: correct the title of paragraph

---

## Round 0.3 · accepted · Accept

Thank you for making the requested modifications to the manuscript.